# Effect of Different Extraction Methods on the Total Phenolics of Sugar Cane Products

**DOI:** 10.3390/molecules28114403

**Published:** 2023-05-28

**Authors:** Azrina Azlan, Sharmin Sultana, Ilya Iryani Mahmod

**Affiliations:** 1Department of Nutrition, Faculty of Medicine and Health Sciences, Universiti Putra Malaysia, Serdang 43400, Malaysia; sharminonti1@gmail.com (S.S.);; 2Research Centre of Excellence for Nutrition and Non-Communicable Diseases, Faculty of Medicine and Health Sciences, Universiti Putra Malaysia, Serdang 43400, Malaysia; 3Laboratory of Halal Science Research, Halal Products Research Institute, Universiti Putra Malaysia, Serdang 43400, Malaysia

**Keywords:** less refined sugar, ethanol extraction, antioxidant, phenolic compounds, α-amylase, α-glucosidase

## Abstract

The health benefits of sugar cane products are attributed to certain antioxidant compounds in plant materials. The presence of antioxidants in plant materials depends on the extraction method in terms of yield and the number of phenolic compounds identified. This study was carried out to evaluate the performance of the three extraction methods, which were selected from previous studies to show the effect of the extraction method on the content of antioxidant compounds in different types of sugar. This study also evaluates the potential of different sugar extracts in anti-diabetic activity based on in vitro assays (α-glucosidase and α-amylase). The results showed that sugar cane extracted with acidified ethanol (1.6 M HCl in 60% ethanol) was the best condition to extract a high yield of phenolic acids compared to other methods. Among the three types of sugar, less refined sugar (LRS) showed the highest yield of phenolic compounds, 57.72 µg/g, compared to brown sugar (BS) and refined sugar (RS) sugar, which were at 42.19 µg/g and 22.06 µg/g, respectively. Whereas, among the sugar cane derivatives, LRS showed minor and BS moderate inhibition towards α-amylase and α-glucosidase activity compared to white sugar (RS). Thus, it is suggested that sugar cane extracted with acidified ethanol (1.6 M HCl in 60% ethanol) is the optimum experimental condition for antioxidant content determination and provides a basis for further exploitation of the health-beneficial resources of the sugarcane products.

## 1. Introduction

Sugarcane is one of the most significant and efficient agricultural products at an industrial level. This crop has the highest production worldwide. According to the Food and Agriculture Organization (FAO) report, about 1.87 billion tonnes of sugarcane were harvested in 2020, and the high production volumes highlight the impact of this sugarcane on the agroindustry [1]. Its impact on the food industry represents tremendous interest due to the remarkable increase in studies about several compounds found in sugar cane products that demonstrate significant nutritive and nutraceutical properties with relevant biological activity both in vivo and in vitro [1,2,3,4]. For instance, sugarcane extracts have been reported to possess anti-proliferative properties against cancer cell lines (leukaemia, stomach, lung, colon, and bladder) [2], protective properties against hepatic damage, anti-thrombotic, and anti-stress properties [3,4], and prevention of hypertension and diabetes disorders [2,3,4,5].

Similarly, sugarcane extracts have also shown antibacterial activity against certain bacteria, such as *Streptococcus mutans* and *Streptococcus sobrinus*, and a few microorganisms responsible for developing dental caries [6]. All the biological activities mentioned above have been related to the presence of certain bioactive compounds found in the sugarcane extract, which are predominantly phenolic acids (such as hydroxycinnamic acid, caffeic acid, and sinapic acid) and polyphenols and flavonoids (such as apigenin, luteolin, and tricin derivatives) [1,7,8,9]. 

Sugarcane is a predominant source for the extraction of bioactive compounds. In sugarcane extract, various phytochemicals with antioxidant properties are present and required to be extracted with little or no effect during extraction [7,10]. Sugars such as refined sugar (RS) and brown sugar (BS) are usually sequentially processed, involving washing, extraction, purification, crystallisation, drying, and packaging stages [7]. Less refined sugar (LRS) is produced from food-grade sugar mills and involves washing, extraction, minimal refining, crystallisation, drying, and packaging [7]. Herein, LRS is not processed as much as RS, and thus retention of essential phytochemicals is higher in LRS than in RS [7]. On the other hand, BS is usually prepared by adding molasses, which determines the grade of sugar (such as higher-grade BS having a strong flavour and darker colour) [10]. Therefore, depending on the compounds present and processing conditions, sugar can be divided into BS, LRS, and RS [7]. 

Extraction is the first step in bioactive compound isolation and is considered one of the essential unit operations. Various extraction methods such as cold pressing, heating reflux, Soxhlet, solid phase, microporous resins, ion exchange, and solvent phase extraction have been widely used to extract bioactive components from natural products [3,4,5,6,7,8,9,10]. However, there are some disadvantages to the existing methods, including uncontrollable low yield, environmental risk and toxicological effects, product degradation, and low product quality, which are frequently observed during extraction [1,5,10]. In recent years, the emergence of green technology, which allows eco-friendly techniques that involve the extraction of phenolic and antioxidant compounds without toxic chemicals, has been used for sugar products [10]. Among these techniques, ultrasonic-assisted extraction [10], solid-phase extraction [1], and ethyl acetate extraction have been proposed to extract bioactive compounds from sugar products because these techniques provide a unique feature to the extracted sample. All three of these techniques gained popularity due to their nontoxic nature and the excellent quality of the extracted compounds, as supported by previous studies for extracting reducing sugars, molasses, and brown sugars [1,8,10].

However, various extraction parameters, such as extraction time, sonication power, and solvent, can directly affect the quality of the extracted samples. Thus, selecting an appropriate extraction method for optimum bioactive compound recovery is necessary. Therefore, this study aimed to apply three different extraction methods (ultrasonic-assisted extraction, solid-phase extraction, and ethyl acetate extraction) to observe the phenolic content variation and yield obtained from the sugar extract. Thus, this work is intended to contribute to the state of the art and could support future research about the characterization, effectiveness, or evaluation of different bioactive molecules from sugarcane products.

In addition, among various chronic diseases where the intake of sugar-based products is a significant concern, diabetes mellitus (DM) is one of them. DM is a metabolic disorder characterised by an abnormally elevated postprandial blood glucose level [11]. Among the different methods for controlling postprandial hyperglycemia, inhibition of α-glucosidase is considered a practical approach. Mammalian α-glucosidase (α-D-glucoside glucohydrolase, EC 3.2.1.20) is the key enzyme that catalyses the hydrolysis of carbohydrates. This enzyme acts by retarding the liberation of glucose from oligosaccharides and disaccharides, reducing postprandial plasma glucose levels [12]. Alpha-amylase inhibitors are remarkably effective in delaying glucose absorption and lowering postprandial blood glucose peaks. Alpha-amylase is one of the major secretory products of the pancreas and salivary glands since it plays a role in the digestion of starch and glycogen [13]. From this point of view, this study also aimed to evaluate the potential of sugar products in anti-diabetic activity based on in vitro assays (α-glucosidase and α-amylase).

## 2. Results and Discussion

### 2.1. Identification and Quantification of Polyphenolic Compounds by HPLC

Thirteen standards were chosen based on previous literature reviews by Payet et al. [8], Barrera et al. [1], and Azlan et al. [7]. Of the 13 standards, only five peaks, namely 5-hydroxymethylfurfural (5-HMF), syringic acid, caffeic acid, p-coumaric acid, and ferulic acid, were detected from the HPLC chromatogram of the sugar extracts. The HPLC chromatograms obtained from the sugar extracts were analysed at a wavelength of 323 nm, and the data are presented in Table 1 and Figure 1. Except for 5-HMF, all the detected peaks were previously reported in the literature. The compound 5-HMF was added based on the principle of the formation of fructose and glucose due to sucrose hydrolysis [14,15]. Usually, 5-HMF is formed from reducing sugars in honey and various processed foods treated in acidic environments or extreme heat conditions due to the Millard reaction [16]. 

The identification of the compounds was confirmed by comparing the retention times of UV-visible spectra with those standards. The major components identified by HPLC were p-coumaric acid, caffeic acid, syringic acid, and ferulic acid. Table 1 shows the phenolic content in different sugar products (LRS, BS, and RS) ranging from 4.48 to 57.72 µg/mL of the extract. The HPLC chromatogram results of the phenolic compounds of the three sugar cane products were analysed at 323 nm, and the results are presented in Figure 1. Nevertheless, 5-HMF can only be identified from the sugar extract, which was prepared using Method A. This method indicates that caffeic acid, p-coumaric acid, ferulic acid, and 5-HMF were detected in LRS, BS, and RS (Figure 1). However, syringic acid was not found in all three types of sugars. This result was inconsistent with a previously published study by Azlan et al. [11], who found the highest amount of syringic acid in minimally refined brown sugar but found it absent in BS and RS samples.

Meanwhile, ferulic acid, syringic acid, and p-coumaric were detected in LRS and BS using solid-phase extraction (Method B). No compound was detected in the RS extract, similar to the RS extract using ethyl acetate extraction (Method C). Caffeic acid, ferulic acid, syringic acid, and p-coumaric acid were observed in the LRS and BS by ethyl acetate extraction. Ferulic acid and p-coumaric acid were the predominant compounds in the sugar samples [8]. Generally, ultrasonic-assisted extraction resulted in the highest phenolic content, followed by ethyl acetate extraction and solid-phase extraction. The ultrasonic-assisted extraction resulted in the LRS and BS extracts being slightly higher in total phenolics than minimally refined brown sugar and brown sugar reported by Azlan et al. [11]. However, the total amount of phenolics in the RS extract was lower than in refined sugar, as reported by Azlan et al. [11]. These discrepancies may be due to differences in the source of sugarcane, cultivation climate, soil type, crop handling method, degree of juice extraction, method of clarification heat treatment of cane juice, and efficiency of impurity removal [17,18,19].

Moreover, the concentrations of inverted sugars in unrefined and less refined sugars also affect the evaluation of phenolic and flavonoid contents [18]. Previous literature also found that brown sugar, where molasses is added, presented the lowest international unit (IU) values due to its deep colour and Millard reaction products [1,11,20,21,22,23]. By contrast, in RS, all the phytochemicals have been strapped off due to extreme boiling and concentration in processing [1,23,24]. 

In this study, the total phenolic content from the three sugar samples varied from as low as 4.48 to 57.72 mg GAE/100 mL depending on the different extraction methods used. According to Chen et al. [10], the ultrasonic extraction method for sugar molasses gives excellent total phenolic contents with about 17.36 mg GAE/100 mL, antioxidant activity of 16.66 mg TE/g, and total anthocyanin content of 31.81 mg/100 g. Another study conducted by Mondol et al. [25] found that ultrasonic extraction is a highly potential and efficient technique to extract RS. This study uses ultrasonication technology to extract the total phenolic compound from reducing sugar. They found total phenolic content ranging between 4.94 and 6.98 mg GAE/g in RS. Our results are similar to the literature [1,8,25], suggesting that utilising a suitable solvent is a critical aspect of the extraction to give the highest polyphenol recovery available in the sugar samples. Using high-polarity solvents, such as ethanol and methanol, might affect the non-polar compounds’ solubility, similar to using low-polarity solvents [19]. Other parameters such as HCl concentration, ethanol concentration, extraction temperature, and time have also significantly impacted the extraction of phenolic compounds from sugar products [1]. 

### 2.2. UHPLC–QTOFMS Analysis 

The sugar extracts from three different extraction methods were further analysed by LCMS analysis. Surprisingly, the number of compounds detected using LCMS in all extracts was higher than the HPLC results. Indeed, phenolic acids, such as caffeic acid and ferulic acid, were found almost in all extracts. According to Payet et al. [8], phenolic acids were the major components of the extracts from sugar products. However, only the LRS and BS extracts applying solid-phase extraction were detected to possess flavonoids (tricin, apigenin, luteolin, and vanillin) (Table 2). This finding shows that the solid-phase extraction (SPE) technique can retain the flavonoid content during the extraction. However, quantitative amounts of the compounds can be lost during solvent evaporation. These results were inconsistent with those documented by Duarte-Almeida et al. [20], who found higher amounts of apigenin, followed by tricin, and, finally, luteolin, in sugarcane juice, molasses, and sugar, respectively, using the SPE method. The differences in LCMS and HPLC results are also due to the differences in the instruments’ sensitivity, accuracy, and specificity. In addition, sensitivity in LCMS allows detection up to picogram concentration [21], but is limited in HPLC due to their limit of detection (LOD) and limit of quantification (LOQ) values. The results of the current study suggested that BS and LRS sugars are composed of phenolic acid and flavonoid compounds, which have been proven to have several health benefits. The results of the present study also reveal that less refined sugar (LRS) and molasses-added sugar (BS) are good sources of phenolic and flavonoid compounds. These results are consistent with the previous studies conducted by Azlan et al. [7], Duarte et al. [20], and Payet et al. [8].

### 2.3. α-Glucosidase and α-Amylase Inhibitory Assay Determination

This study also evaluated α-amylase and α-glucosidase inhibitory activities of these three sugar types. Briefly, the α-glucosidase inhibitory activity was designed to assess the ability of the test sample to suppress the active enzyme from catalysing the conversion of glucose from the disaccharides, which occurs in the small intestine [14]. In this study, PNPG, a specific substrate that allows hydrolysation to 4-nitrophenol (a yellow-coloured product) by α-glucosidase enzyme, was quantitated at a maximum wavelength of 405 nm. Similarly, α-amylase inhibitory activity was used to measure the free carbonyl group of the reducing sugar (maltose) converted by the α-amylase enzyme from the complex carbohydrates (potato starch). The aldehyde group from the maltose reduces the yellow-coloured DNS to form 3-amino-5-nitrosalicylic acid (a brick-red-coloured solution) [22]. The concentration of maltose in the sample was determined at a wavelength of 540 nm. Table 3 demonstrates the percentage inhibition of the α-amylase and α-glucosidase activities of the sugar samples. The LRS and BS show minor inhibition in the α-amylase assay. However, no inhibition was detected in the RS sample for both assays. By contrast, in the α-glucosidase assay, LRS and BS exhibited inhibitory activities of 25.16 and 21.22, respectively, at 100 mg/mL. These activities can be attributed to the high presence of caffeic acid and ferulic acid, which have been reported as α-glucosidase inhibitors [23]. 

## 3. Materials and Methods

### 3.1. Sample Extraction

Among the various studies, three types of extraction methods based on previous research findings [1,8,10] such as ultrasonic-assisted extraction [10], solid-phase extraction [1], and ethyl acetate extraction [8], were selected for three types of sugar products (LRS, BS, and RS), to observe the effect of extraction methods on phenolic content variation and yield recovery from the sugar cane extracts. Here, LRS is not processed as much as RS, and thus the retention of essential phytochemicals is supposed to be higher in LRS than in RS [7]. By contrast, BS is usually prepared by adding molasses, which gives it a brown colour. LRS, BS, and RS were purchased in triplicate from grocery stores of a particular brand in Malaysia. All sugar samples (3.0 kg each) were stored at room temperature (25 °C) until extraction. All chemicals, standards, and solvents of analytical grade were purchased from the Sigma-Aldrich Company (Germany) through the authorised agent in Malaysia.

### 3.2. Ultrasonic-Assisted Extraction (Method A)

Ultrasonic-assisted extraction (UAE) was selected from the previous study conducted by Chen et al. [10]. Their study uses response surface methodology to optimise experimental conditions for the UAE of some bioactive compounds from sugar beet molasses. We have applied this method in our study with slight modifications. Briefly, 60 g of powdered sugar samples (RS, BS, and LRS) were dissolved in 600 mL of acidified ethanol (1.6 M HCl in 60% ethanol). The mixture was then sonicated for 30 min at a controlled temperature of 30–33 °C. The mixture was then concentrated under a vacuum and freeze-dried until further use.

### 3.3. Solid-Phase Extraction (Method B)

Solid-phase extraction was selected from the previous study with slight modifications, as conducted by Barrera et al. [1]. We chose this extraction method because they used RS and BS for their studies. Firstly, sugar products were dissolved in bio-distilled water with a 1:3 *w*/*v* ratio and centrifuged at 3500 rpm for 10 min. The supernatant after centrifugation was collected for further analysis. Then, solid-phase extraction was performed using polyamide columns (CHROMABOND) previously conditioned with 10 mL of methanol and 30 mL of bidistilled water. Five mL aliquots of the extracts were fractionated in the polyamide columns and further washed with 10 mL of bidistilled water. Then the mixture was eluted with 25 mL of methanol and 25 mL of methanol/ammonia (99.5:0.5 *v*/*v*). Finally, the volume extracted (50 mL) was evaporated to dryness at 40 °C under vacuum conditions in a rotary evaporator.

### 3.4. Ethyl Acetate Extraction (Method C)

Ethyl acetate extraction was selected from the previous study by Payet et al. [8]. Their study extracted polyphenols from seven sugar product categories with ethyl acetate [8]. We have applied this method in our study with slight modifications. First, approximately 0.3 g of each sugar product was dissolved in 2 mL of ethyl acetate. The solution was then extracted with sodium hydroxide (2 × 1 mL; 10%, *w*/*w*) using a vortex mixer and centrifugation (at 4000 rpm for approximately 5 min). The pH of the resulting aqueous layer was also adjusted to nearly 4 by adding 5 N hydrochloric acid and extracting with ethyl acetate (3 × 2 mL). Finally, the combined organic layer was dried over anhydrous sodium sulphate, concentrated to 0.2 mL using a rotary evaporator (45 °C, 90 mbar), and dried with liquid nitrogen gas.

### 3.5. High-Performance Liquid Chromatography (HPLC) Analysis

Total phenolic compounds from three different types of sugar were identified and quantified using the analytical reversed-phase HPLC method (HPLC Agilent 1100 series Agilent Technologies, Berlin, Germany) equipped with an automatic sampler, degasser, binary pump, diode-array detector (DAD), and reversed-phase column, Luna C18 (250 × 4.6 mm ID; particle size 5 μm) maintained at 40 °C. 

Briefly, the UV was recorded in three channels (254 nm, 280 nm, and 360 nm), and the wavelength of 280 nm was selected for quantification. The following elution solvents were used: (A) water/tetrahydrofuran/trifluoroacetic acid (98:2:0.1) and (B) acetonitrile. The solvent gradient was similar to that of Barrera et al. [1]. Determinations were performed in triplicates, with 20 µL being the volume injected. Identification followed a comparison of UV spectra at the selected wavelength and retention times with standards, and quantification was based on external calibration. There were 13 standards used for this study. The selection of standards followed some previous literature reviews [1,7,8]. 

### 3.6. UHPLC–QTOFMS Analysis

The phenolic compound analysis was further performed on the Agilent 1290 Infinity LC system coupled to the Agilent 6520 Accurate-Mass Q-TOF mass spectrometer with a dual ESI source consisting of a binary pump, a vacuum degasser, an auto-sampler, and a column oven. Phenolic compounds were chromatographically separated using a column Agilent ZORBAX Eclipse XDB-C18 Analytical (150 mm × 4.6 mm, 5-micron), maintained at 40 °C. A linear binary gradient of water (0.1% formic acid) and methanol was used as mobile phases A and B, respectively. The composition of the mobile phase was changed during the run as follows: The mobile phase composition was changed during the run as follows: 0 min, 30% B; 0.5 min, 70% B; 95.00 min, 100% B; 20.00 min, 1% B. The flow rate was set to 0.7 mL/min, and the injection volume was 1 μL.

The UHPLC system was coupled to a Mass Q-TOF mass spectrometer with a dual ESI source spectrometer from Agilent. The ion source was operated in negative electrospray ionisation (ESI) mode under the following specific conditions: capillary voltage, 1.50 kV; reference capillary voltage, 3.50 kV; source temperature, 120 °C; desolvation gas temperature, 300 °C; and cone gas flow, 10 L/h. Nitrogen (>99.5%) was employed as desolvation and cone gas. Data were acquired in high-definition MSE (HDMSE) mode in the m/z range of 100–3200 at 1.02 spectra/s. In addition, bioactive compounds were identified and quantified using the polyphenol library in the Metlink Database. 

### 3.7. α-Amylase Inhibitory Assay

The α-amylase inhibition activity assay was carried out using the demonstrated method with modifications [14]. In a 96-well plate, a reaction mixture containing 50 µL of 100 mM phosphate buffer at pH 6.8, 10 µL of 2 U/mL porcine α-amylase, and 20 µL of varying concentrations of the extract (6.13, 12.5, 25, 50, and 100 mg/mL) were incubated at 37 °C for 20 min. Then, 20 µL of a substrate containing 1% starch dissolved in 100 mM phosphate buffer at pH 6.8 was added, and the mixture was further incubated at 37 °C for about 30 min. The reaction was then stopped by adding 100 µL of the DNS colour reagent and boiling for 10 min. The absorbance was measured using a spectrophotometer at 540 nm wavelength. The α-amylase inhibitory activity was calculated using the equation: [(An − As)/An] × 100%, where An is the difference in absorbance of the negative control and all the blanks, and As is the difference in absorbance of the sample and all the blanks. The α-glucosidase inhibitory activity was expressed as an IC_50_ value (mg/mL) to represent the sugar concentration needed to inhibit enzyme activity by 50%. In this study, acarbose was tested and used as a positive control.

### 3.8. α-Glucosidase Inhibitory Assay

The α-glucosidase inhibitory assay was performed in a 96-well plate according to the previously published studies [14] with slight modifications. Firstly, 10 μL of the test compound in 0.5% DMSO was diluted with 100 μL buffer (0.03 M phosphate buffer pH 6.5) and mixed with 15 μL of the enzyme (0.2 U/mL) solution and incubated for 10 min at room temperature. Then, 75 μL of the substrate containing *p*-nitrophenyl α-D-glucopyranoside (0.5 mM concentration in 0.05 M phosphate buffer, pH 6.5) was added to the mixture and allowed to incubate at room temperature for 15 min. Afterward, the reaction was terminated by adding 50 μL of glycine (2 M, pH 10). The optical density (OD) was determined by spectrophotometry at approximately a length of 405 nm. Finally, the percent inhibition of the enzyme was calculated compared with the control and expressed as the mean ± SD. The α-glucosidase inhibitory activity was calculated using the equation given below.
[(An − As)/An] × 100%
where An is the difference in absorbance between the negative control and all the blanks, and As is the difference in absorbance between the sample and all the blanks. In this study, acarbose was also tested and used as a positive control.

## 4. Conclusions

The health benefits of sugar products are dependent on the presence of antioxidants in plant materials. The amount of antioxidants found in the products further depends on the extraction method in terms of yield. This study aims at the methodological improvement of the extraction process of sugar phenolic compounds to ensure maximum health benefits. This study found that different sugars have shown mixed results in the content of phenolic compounds due to their processing conditions. Among the three types of sugar, LRS showed the highest yield in phenolic compounds compared to BS and RS sugar because the minimal refining process of LRS retained some of its phytochemicals, including phenolic acid content. However, LRS exhibited minor inhibition of α-amylase and moderate inhibition of α-glucosidase activity.

On the other hand, among the three proposed methods (A, B, and C), sugar cane extracted with acidified ethanol (1.6 M HCl in 60% ethanol) (method A) was revealed to be the preferred condition to extract a high yield of phenolic acids compared to other methods. This extraction method exhibited adequate performance, and the bioactive compounds extracted from sugar were not significantly reduced. Thus, this study suggests that acidified ethanol (1.6 M HCl in 60% ethanol) is the best extraction method for bioactive compound determination for sugar samples. However, further detailed investigations are required to evaluate the usefulness of this method with other plant materials containing sugar derivatives.

## Figures and Tables

**Figure 1 molecules-28-04403-f001:**
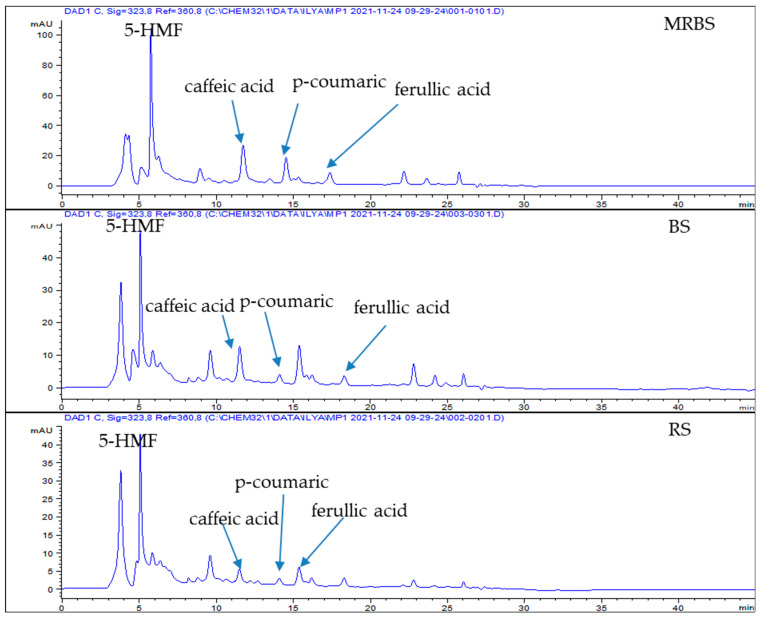
HPLC chromatograms of the sugar cane extracts obtained from acidified ethanol extraction by elution with methanol and methanol/ammonia, monitored at 323 nm.

**Table 1 molecules-28-04403-t001:** Phenolic compounds in non-refined cane sugars determined by HPLC.

Compounds	Retention Time	Ultrasonic-Assisted Extraction (A)			Solid-Phase Extraction (B)			Ethyl Acetate Extraction (C)		
		LRS	BS	RS	LRS	BS	RS	LRS	BS	RS
5-HMF	5.858	14.76	6.9	5.54	ND	ND	ND	ND	ND	ND
Syringic acid	10.632	ND	ND	ND	Trace	Trace	ND	Trace	Trace	ND
Caffeic acid	11.247	24.03	15.53	8.4	ND	ND	ND	Trace	Trace	ND
p-coumaric acid	14.414	15.68	2.36	1.56	9.9	4.24	ND	18.63	10.61	ND
Ferulic acid	15.135	3.25	17.4	6.7	1.26	0.24	ND	7.36	5.6	ND
	TOTAL	57.72	42.19	22.06	11.16	4.48		25.99	16.21	

Concentrations are expressed in µg/g sugar samples. ND—Not detected in the chromatogram. Trace—(<1.0 μg/g sample below the quantification limit).

**Table 2 molecules-28-04403-t002:** LCMS identification of sugar extract content.

Compounds	RT(min)	[M-H]^−^	Ultrasonic-Assisted Extraction (A)			Solid-Phase Extraction (B)			Ethyl Acetate Extraction (C)		
			LRS	BS	RS	LRS	BS	RS	LRS	BS	RS
Phenolic acids											
Syringic acid	1.93	197	*	*	*	*	*	-	*	*	-
Caffeic acid	2.07	179	*	*	*	*	*	*	*	*	*
p-coumaric acid	20.76	163	*	*	*	*	*	*	*	*	-
Ferulic acid	21.49	193	*	*	*	*	*	*	*	*	*
Chlorogenic acid	42.89	353	*	*	*	*	*	*	-	-	-
3,4-hydroxybenzoic acid	11.13	153	*	*	*	*	*	*	-	-	-
Vanilic acid	16.86	167	*	*	*	*	*	*	-	-	-
Flavonoids											
Tricin	32.37	329	-	-	-	*	*	-	-	-	-
Apigenin	22.90	269	-	-	-	*	*	-	-	-	-
Luteolin	35.23	285	-	-	-	*	*	-	-	-	-
Vanillin	15.53	151	-	-	-	*	*	-	-	-	-

* Indicates compound detected in the LCMS.

**Table 3 molecules-28-04403-t003:** α-Amylase and α-Glucosidase inhibitory assays of sugar samples at 100 mg/mL.

Sample	Inhibition (%) α-Amylase	Inhibition (%) α-Glucosidase
LRS	4.12 ± 0.70 ^a^	25.16 ± 0.80 ^c^
BS	4.51 ± 0.26 ^b^	21.22 ± 0.51 ^d^
RS	No inhibition	No inhibition
Acarbose (IC_50_)	0.40 ± 0.21 µg/mL	2.25 ± 0.63 µg/mL

Different letters indicate a significant difference at *p* < 0.05 among the samples.

## Data Availability

Not applicable.

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
