# Peer review of "Effect of Different Extraction Methods on the Total Phenolics of Sugar Cane Products"

_molecules, 2023, doi:10.3390/molecules28114403_

Round 1

Reviewer 1 Report

Dear Authors,

Your study is focusing on a very important product - sugar, and a methodological improvement of the extraction of its phenolic compounds. I divide my review into on two parts – first: Contents, second: Formal mistakes.

1.      First of all, in the Material and Method part I miss the description of the sugar samples. What does it mean – Less refined sugar? It is a white? Brown? What is the difference between LRS and MRBS  - minimally refined brown sugar (that is mentioned in other articles, moreover also in this article in the conclusion part you used MRBS and not LRS word – why?). Where the sugar samples were come from?

For the identification and quantification of the phenolic compounds you used HPLC and LC-MS. In the second case only the presence was given in Table 2, without any parameters giving on the amount. Please give some proper amounts as well, otherwise Table 2 cannot be used.

The Conclusion part is rather short, only one paragraph. In this chapter you should evaluate your results again compared to the literature data emphasizing the novelty of your study, highlighting the most important findings. The first sentence it too common: “this study investigated the effect of three different extraction methods on total phenolic content” – of what? Why it was important to compare these methods?

2.      Formal mistakes:

Row 17-18: letter size is too small.

In the introduction part you use “less refined sugar” (LRS) and at the end of the 62. row MRBS (without giving the explanation of this word). Please clarify what is the difference between them? Or please use the same name, if these are synonyms.

Results and Discussion, row 108.  – HMF, please write directly before the molecule name (now it is found later, row, 111). Complete the sentence starts in 109 row: “The HPLC chromatograms….data are presented in Table 1 and Figure 1. In this case the sentence starts in the 122 row can be deleted (“The HPLC chromatogram results….).

Method A, B and C should be indicated in the table and in the figure as well.

It is confusing, that the order of the sugar products are different – in Figure 1: LRS, BS, RS; in Table 1: LRS, RS, BS – please put the same order in each case.

Table 2. Letter size is too big.

Chapter 2.3. The first sentence is too long, and complicated to understand. Divide it into two parts. Row 209: PNPG: explain!

Chapter 3.6 Row 288: “The mobile phase composition was changed during the run as follows:” – duplicate, delete the first one!

Chapter 3.8. Row 333.: you wrote agarose – correct it to Acarbose!

Conclusions: in the abstract and in your results you used LRS, however in the Conclusions you highlighted MRBS sample having the highest yield of phenolic compounds - row 342. It is a serious mistake - please correct or explain!

Author Contributions: delete, if it is not necessary!

Author Response

Reviewer report 1:

Your study is focusing on a very important product - sugar, and a methodological improvement of the extraction of its phenolic compounds. I divide my review into on two parts – first: Contents, second: Formal mistakes.

Author response:Thanks to the reviewers for critical reading of the manuscript, providing valuable comments and suggestions to modify our manuscript substantially. Please find attached herewith the responses to the reviewers’ comments point by point on the paper entitled“ Effect of different extraction methods on the total phenolics of sugar cane products”. Please note that all the amendments in the revised manuscript (MS) are marked up with red color text.

Q1. First of all, in the Material and Method part I miss the description of the sugar samples. What does it mean – Less refined sugar? It is a white? Brown? What is the difference between LRS and MRBS - minimally refined brown sugar (that is mentioned in other articles, moreover also in this article in the conclusion part you used MRBS and not LRS word – why?). Where the sugar samples were come from?

Author response: Description about the three types of sugar were given in the 2.1 section. MRBS and LRS are both less refined sugar products. In the manuscript, I have changed MRBS into LRS. (Here, LRS is not processed as much as RS, and thus the retention of essential phytochemicals is supposed to be higher in LRS than RS [7]. By contrast, BS is usually prepared by adding molasses that gives BS brown colour. LRS, BS and RS were purchased in triplicate from grocery stores of a particular brand in Malaysia. All sugar samples (3.0 kg each) were stored at room temperature (25°C) until extraction. All chemicals, standards, and solvents of analytical grade were purchased from Sigma-Aldrich Company (Germany) through the authorized agent in Malaysia)

Q2. For the identification and quantification of the phenolic compounds you used HPLC and LC-MS. In the second case only the presence was given in Table 2, without any parameters giving on the amount. Please give some proper amounts as well, otherwise Table 2 cannot be used.

Author response: The amount found in the LCMS method was very low. Therefore, we have presented the data as detectable. I hope honourable reviewer will understand. Thank you

Q3. The Conclusion part is rather short, only one paragraph. In this chapter you should evaluate your results again compared to the literature data emphasizing the novelty of your study, highlighting the most important findings. The first sentence it too common: “this study investigated the effect of three different extraction methods on total phenolic content” – of what? Why it was important to compare these methods?

Author response: Conclusion part has been revised as suggested by reviewers.

  1. 2.      Formal mistakes:

Q1 Row 17-18: letter size is too small.

Author response: Revised.

Q2 In the introduction part you use “less refined sugar” (LRS) and at the end of the 62. row MRBS (without giving the explanation of this word). Please clarify what is the difference between them? Or please use the same name, if these are synonyms.

Author response: MRBS has been re-write as LRS. Thank you

Q3 Results and Discussion, row 108.  – HMF, please write directly before the molecule name (now it is found later, row, 111). Complete the sentence starts in 109 row: “The HPLC chromatograms….data are presented in Table 1 and Figure 1. In this case the sentence starts in the 122 row can be deleted (“The HPLC chromatogram results….).

Author response: Revised.

Q4 Method A, B and C should be indicated in the table and in the figure as well.

Author response: Indicated in the table and figure. Thank you

Q5 It is confusing, that the order of the sugar products are different – in Figure 1: LRS, BS, RS; in Table 1: LRS, RS, BS – please put the same order in each case.

Author response: Revised table 1 and 2 to put the same order.

Q6 Table 2. Letter size is too big.

Author response: Revised.

Q7 Chapter 2.3. The first sentence is too long, and complicated to understand. Divide it into two parts. Row 209: PNPG: explain!

Author response: Revised. Thank you

Q8 Chapter 3.6 Row 288: “The mobile phase composition was changed during the run as follows:” – duplicate, delete the first one!

Author response: Deleted.

Q9 Chapter 3.8. Row 333.: you wrote agarose – correct it to Acarbose!

Author response: Corrected. Thank you

Q10 Conclusions: in the abstract and in your results, you used LRS, however in the Conclusions you highlighted MRBS sample having the highest yield of phenolic compounds - row 342. It is a serious mistake - please correct or explain!

Author response: Corrected. Thank you

Q11 Author Contributions: delete, if it is not necessary!

Author response: Parts of Journal guidelines. Therefore, I would like to keep in manuscript.

Reviewer 2 Report

The subject of the article was the analysis of phenolics in sugar cane products. The authors performed three extraction methods and then determined the content of phenolic acids. They also performed a qualitative study of the composition and assessed the hypoglycemic potential to reduce the blood sugar level. The measurement was carried out correctly. The text is written in correct English, the structure is simple and clear. However, the article does not contain novelty.  Studies on the composition of sugar industry products are pulished in large numbers, mostly concerning molasses and other by-products that have a richer composition than more or less refined sugar:

https://doi.org/10.1016/j.foodchem.2010.09.059. https://doi.org/10.3390/foods11244025

The obtained results do not bring anything new to the research on sugar cane products. The publication requires many changes and deep rethinking. At this stage I recommend rejecting it.

Other comments:

-       - Lack of any details regarding the analyzed samples - source/origin, sugar cane plant varieties, basic characteristics of the samples;

-          - No indication of other ingredients - the content of sugar, amino acids, vitamins (mainly vitamin B6, present in molasses) and minerals (K, Ca, Mg, Fe, Mn);

-          - MRBS - no explanation of the abbreviation;

-          - Less refined sugar (LRS) - what does "less" mean? - especially in the context of the largest amount of phenolic acids?

-          - Abbreviations should be explained in the text, not just in the abstract;

-          - Table 1. "Phenolic compounds in unrefined cane sugars determined by HPLC" – the table contains also results for the RS "refined sugar” sample;

-         - line 204 - "Hypoglycemic potential of three sugar samples to lower blood sugar" I don't see the sense of these studies with respect to sugar. How will the authors explain the ability of BS/LRS to decrease the sugar level in blood?

-          - What was the purpose of ethyl acetate extraction, since the composition of polyphenols is known, and this solvent is not the best for their extraction.

Author Response

Reviewer 2

The subject of the article was the analysis of phenolic in sugar cane products. The authors performed three extraction methods and then determined the content of phenolic acids. They also performed a qualitative study of the composition and assessed the hypoglycemic potential to reduce the blood sugar level. The measurement was carried out correctly. The text is written in correct English, the structure is simple and clear. However, the article does not contain novelty.  Studies on the composition of sugar industry products are published in large numbers, mostly concerning molasses and other by-products that have a richer composition than more or less refined sugar:

https://doi.org/10.1016/j.foodchem.2010.09.059. https://doi.org/10.3390/foods11244025

Author comments: Thanks to the reviewers for critical reading of the manuscript, providing valuable comments and suggestions to modify our manuscript substantially. Please find attached herewith the responses to the reviewers’ comments point by point on the paper entitled“ Effect of different extraction methods on the total phenolics of sugar cane products”. Please note that all the amendments in the revised manuscript (MS) are marked up with red color text.

Q1. Lack of any details regarding the analyzed samples - source/origin, sugar cane plant varieties, basic characteristics of the samples;

Author response: We have described sugar sample in the section 2.1.

 Q2. No indication of other ingredients - the content of sugar, amino acids, vitamins (mainly vitamin B6, present in molasses) and minerals (K, Ca, Mg, Fe, Mn);

Author response: This study focused only on the methodological improvement of the extraction of sugar phenolic compounds. Therefore, author think that the above information not relevant with the purpose of this study.

 Q3. MRBS - no explanation of the abbreviation.

Author response: MRBS and LRS are both less refined sugar products. MRBS has been changed into LRS

Q4 Less refined sugar (LRS) - what does "less" mean? - Especially in the context of the largest amount of phenolic acids?

Author response: Described already in the Introduction part and section 2.1.

Q5 Abbreviations should be explained in the text, not just in the abstract.

Author response: Thank you so much for your informative comments. Revised abbreviations in the text.

Q6 Table 1. "Phenolic compounds in unrefined cane sugars determined by HPLC" – the table contains also results for the RS "refined sugar” sample;

Author response: We have kept RS sugar to compare unrefined sugar.

Q7. line 204 - "Hypoglycemic potential of three sugar samples to lower blood sugar" I don't see the sense of these studies with respect to sugar. How will the authors explain the ability of BS/LRS to decrease the sugar level in blood?

Author response: We have deleted this sentences and revised the 1st sentence in the section 3.3. Thank you.

What was the purpose of ethyl acetate extraction, since the composition of polyphenols is known, and this solvent is not the best for their extraction.

Author response: The aim of this study is the methodological improvement of the extraction of phenolic compounds in sugar sample. We have used ethyl acetate extraction method as this method has been previously documented by other author for sugar sample extraction (Payet, B.; Sing, A.S.C.; Smadja, J.  Comparison of the concentrations of phenolic constituents in cane sugar manufacturing products with their antioxidant activities, J. Agric. Food Chem. 54, 7270–7276. https://doi.org/10.1021/jf060808o). Our study aims was to compare three method those as been previously used for sugar research. For this reason, we used ethyl acetate extraction techniques in our research.  

Round 2

Reviewer 1 Report

Dear Authors, I have no further comments on your paper. 

Author Response

Dear Authors, I have no further comments on your paper. 

Author response: Thank you very much.

Reviewer 2 Report

The revised version requires following corrections:

- The authors did not elaborate on the novelty part of the paper.

- They did not add information regarding any details of the analyzed samples - source/origin, sugar cane plant varieties, basic characteristics of the samples – only that they bought them (what exactly?) in the shop.

- Table 1 - title should be “Phenolic compounds in the cane sugars determined by HPLC”.

- Table 1 also lacks standard deviations of the obtained values and basic statistics (e.g., Tukey's range test).

Author Response

Reviewer report 2:

The revised version requires following corrections:

Author response: Thanks to the reviewers for critical reading of the manuscript, providing valuable comments and suggestions to modify our manuscript substantially. Please find attached herewith the responses to the reviewers’ comments point by point on the paper entitled“ Effect of different extraction methods on the total phenolics of sugar cane products”. Please note that all the amendments in the revised manuscript (MS) are marked up with red color text.

Q1. The authors did not elaborate on the novelty part of the paper.

Author response: Revised. Please see the newly track changes part. Thank you

Q2. They did not add information regarding any details of the analyzed samples - source/origin, sugar cane plant varieties, basic characteristics of the samples – only that they bought them (what exactly?) in the shop.

Author response: Please see section 2.1 (LRS, BS and RS were purchased in triplicate from grocery stores of a particular brand in Malaysia. All the sugar tested in this study were manufactured by Malaysian local companies and produced from sugar cane plant. We bought each sugar in 1 kg package. BS is usually prepared by adding molasses and having a strong flavor and darker color while LRS is a less refined sugar and having light brownish color. By contrast, RS is a fully refined cane sugar with white crystal color)

Q3. Table 1 - title should be “Phenolic compounds in the cane sugars determined by HPLC”.

Author response: Done

Q4. Table 1 also lacks standard deviations of the obtained values and basic statistics (e.g., Tukey's range test).

Author response: Revised and analysed data in Table 1 as per honourable reviewer suggestions. Please see table 1. Thank you.

Round 3

Reviewer 2 Report

I recommend accepting the article in present form.